# Efficient Photodynamic Therapy of Prostate Cancer Cells through an Improved Targeting of the Cation-Independent Mannose 6-Phosphate Receptor

**DOI:** 10.3390/ijms20112809

**Published:** 2019-06-08

**Authors:** Elise Bouffard, Chiara Mauriello Jimenez, Khaled El Cheikh, Marie Maynadier, Ilaria Basile, Laurence Raehm, Christophe Nguyen, Magali Gary-Bobo, Marcel Garcia, Jean-Olivier Durand, Alain Morère

**Affiliations:** 1IBMM, Univ Montpellier, CNRS, ENSCM, 34093 Montpellier, France; elise.bouffard@gmail.com (E.B.); christophe.nguyen@umontpellier.fr (C.N.); m.garcia@nanomedsyn.com (M.G.); 2ICGM, Univ Montpellier, CNRS, ENSCM, 34095 Montpellier, France; chiaramauriellojimenez@gmail.com (C.M.J.); laurence.raehm@umontpellier.fr (L.R.); 3NanoMedSyn, 34093 Montpellier, France; k.elcheikh@nanomedsyn.com (K.E.C.); m.maynadier@nanomedsyn.com (M.M.); i.basile@nanomedsyn.com (I.B.)

**Keywords:** mannose 6-phosphate analogues, binding affinity, mesoporous silica nanoparticles, photodynamic therapy

## Abstract

The aim of the present work is the development of highly efficient targeting molecules to specifically address mesoporous silica nanoparticles (MSNs) designed for the photodynamic therapy (PDT) of prostate cancer. We chose the strategy to develop a novel compound that allows the improvement of the targeting of the cation-independent mannose 6-phosphate receptor, which is overexpressed in prostate cancer. This original sugar, a dimannoside-carboxylate (M6C-Man) grafted on the surface of MSN for PDT applications, leads to a higher endocytosis and thus increases the efficacy of MSNs.

## 1. Introduction

The main issues in cancer treatment are the emergence of resistance and a lack of specificity of the cytotoxic compounds, leading to severe side effects. To overcome the latter drawback, researchers have focused on the development of targeted therapies that would distinguish cancer cells from healthy cells [1,2,3]. In the aim of creating efficient personalized therapies, the use of specific biomarkers is of primary importance. In this field, lectins represent a very large receptor family whose interactions with specific carbohydrates play a key role in bio-recognition processes [4,5]. Combined with nanomedicine, the specific membrane targeting of lectins, which are overexpressed in tumors with carbohydrate ligands, is an important area of research [6,7,8,9,10]. This approach has recently led to clinical trials, such as the targeting of asialoglycoprotein receptors in hepatocytes using copolymer–doxorubicin–galactosamine conjugates for the treatment of liver cancer [11]. Indeed, the multivalence brought about by the conjugation of carbohydrates to nanoparticles leads to a clustering effect that allows an efficient interaction with the targeted lectin and an internalization of the nano-conjugate in cancer cells through receptor-mediated endocytosis [12,13].

For the treatment of prostate cancer, the cation-independent mannose 6-phosphate receptor (CI-M6PR) appears as an interesting target as it is overexpressed in 84% of prostate cancers [14]. Furthermore, since one of the roles of the CI-M6PR is the endocytosis of proteins bearing M6P ligands [15] and their addressing to the lysosomes, this receptor can be used to enter a drug inside cells. It has been demonstrated that the replacement of the phosphate moiety of M6P by isosteric groups, such as carboxylate, phosphonate, or malonate, can avoid the phosphate cleavage by phosphatases while keeping a good affinity for the CI-M6PR [16,17,18,19,20]. We therefore synthesized in a previous work a carboxylate analogue of M6P and anchored it on the surface of mesoporous silica nanoparticles (MSNs) functionalized for an efficient photodynamic therapy (PDT) of prostate cancer [14].

To further enhance the efficiency of MSN, we anticipated that a dimannoside derivative would be more active than the monosaccharide. Indeed, the M6P moieties are present on the mannose-rich oligosaccharide chains of lysosomal enzymes and the M6P residue is linked to a mannose residue through an α-1,2-linkage [21]. Herein, we describe the synthesis of a dimannoside-carboxylate (M6C-Man), a mimic of the M6P-α-1,2-Man, functionalized with an ethyl squarate linker that allows its anchoring on the surface of MSN designed for PDT applications. The synthesis was successfully performed in 16 steps. In order to evaluate this dimannoside analogue, the monosaccharide carboxylate (M6C) bearing the same linker was also prepared and anchored on the same MSN as for the dimannoside analogue. The efficiency of the endocytosis of the two MSNs in prostate cancer cells was monitored by near-infrared fluorescence and the results of PDT were compared.

## 2. Results and Discussion

### 2.1. Synthesis of the M6C **5** and M6C-Man **17** Saccharidic Ligands

The M6C **5** and M6C-Man **17** derivatives were synthetized following the synthetic pathways described in Scheme 1 and Scheme 2. One of the main steps for the synthesis of these saccharidic ligands was the introduction of the carboxylate function on the side chain of the mannose moieties. This functionalization was realized in both cases by a Dess–Martin oxidation of the alcohol at the 6-position followed by a Horner–Wadsworth–Emmons reaction using the triethyl phosphonacetate anion. This way, the unsaturated carboxylates **3** and **14** were obtained respectively for the mono and dimannoside series (Scheme 1 and Scheme 2). Another critical step was the functionalization of the aglycone moieties with the diethyl squarate. This reaction was performed on the fully deprotected monosaccharide **4** and disaccharide **16** to provide the M6C **5** and M6C-Man **17,** ready to be grafted onto the surface of the nanoparticles, respectively. All syntheses and characterizations are given in the Appendix A.

### 2.2. Affinity for CI-M6PR

The affinity of the M6P analogues for the biotinylated CI-M6PR was measured according to a previously described procedure [16]. To avoid any reaction of the squarate moiety with amine functions of the receptor, this binding assay was performed with the compounds **4** and **16**. This assay demonstrated that the disaccharide **16** has a 5.7-fold higher affinity for the CI-M6PR (IC_50_ = 0.28 µM) than the monosaccharide **4** (IC_50_ = 1.6 µM). This result is in accordance with literature data. Indeed, M6P derivatives linked to a second mannose by an α-1,2-linkage are known to lead to about 3-fold higher affinity for the CI-M6PR [22].

The M6C and M6C-Man derivatives were then grafted on mesoporous silica nanoparticles (MSNs) designed for the monophotonic PDT.

### 2.3. Synthesis of Mesoporous Silica Nanoparticles

The MSNs were synthesized according to the procedure described by Brevet et al. [23] consisting of a co-condensation of the photosensitizer with tetraethoxysilane in the presence of cetyltrimethylammonium bromide and in basic media. We used a neutral porphyrin (POR) bearing a maleimide arm [24], which was functionalized with 3-mercaptopropyltriethoxysilane (Scheme 3a).

Following this procedure, MSNs were obtained presenting a 130 nm diameter, confirmed by transmission electron microscopy (TEM) and by Gaussian analysis (Appendix A). The size and the porosity of MSNs make them suitable nanocarriers for nanomedicine due to their high surface area of 1109 m^2^·g^−1^ and the 2 nm pore diameter (Appendix A).

Then, MSNs were diluted in EtOH and submitted to UV–visible spectroscopy in order to verify the successful encapsulation of the porphyrin. As shown in Appendix A, the absorption of the Soret band at 415 nm and the four Q bands at the following wavelengths: 520, 556, 593, and 650 nm confirm the presence of porphyrin in the MSN. The quantitative analysis of the Soret band has allowed for the extrapolation of the quantity of porphyrin encapsulated in MSN. A loading of 11 µmol of porphyrin per gram of nanoparticles was obtained.

The surface of MSN (Scheme 3b) was first functionalized with aminopropyltriethoxysilane (APTES). Amino groups were then reacted with M6C or M6C-Man to give MSN-M6C and MSN-M6C-Man, respectively. The surface functionalization with the different ligands was confirmed by UV–Vis analysis (Appendix A). The quantity of grafted carboxylate derivatives was measured by UV/vis spectroscopy and 352 µmol·g^−1^ for MSN-M6C and 329 µmol·g^−1^ for MSN-M6C-Man were obtained. This same order of magnitude of grafting allows us to compare the behavior of the MSN in prostate cancer cells.

### 2.4. In Vitro PDT Experiments

These nanoparticles were used for in vitro PDT experiments on human prostate adenocarcinoma cells LNCaP. The cells were incubated with MSN, MSN-M6C, or MSN-M6C-Man at a concentration of 80 µg·mL^−1^. After different incubation times (3 h, 6 h, 9 h, or 18 h), the cells were irradiated for 20 min with a laser at 650 nm (3 mW, 11.25 J·cm^−2^) (Figure 1). The benefit of the nanoparticles’ functionalization with the M6C-Man (versus the M6C and the bare MSN) was confirmed with lower cell survival rates for every incubation time, the most significant difference being for a 6 h incubation with only 27% of cell survival for MSN-M6C-Man and 65% of cell survival with MSN-M6C.

This experiment thus demonstrated that the use of MSN-M6C-Man not only increases the cell death but is also more efficient in a short period of treatment. This higher efficacy is consistent with the higher affinity for CI-M6PR of MSN-M6C-Man compared to MSN-M6C.

The endocytosis of the different MSNs was followed with confocal microscopy merged images in LNCaP prostate cancer cells (Figure 2). The cells were incubated with the nanoparticles for 24 h. In comparison to bare MSN, both MSN-M6C-Man and MSN-M6C were highly internalized. The efficiency of the targeting for prostate cancer is clearly visible by fluorescence imaging at a wavelength of 633 nm.

The experiments presented in Figure 1 and Figure 2 demonstrated a better therapeutic effect of M6C-Man-MSN, suggesting the strong advantage of the targeting of CI-M6PR to focus on for treating prostate cancer. 

### 2.5. Endocytosis Mediated by CI-M6PR

Figure 3 shows the involvement of CI-M6PR in the targeting and PDT efficiency of cancer cells by M6C-Man-MSN. In Figure 3a, a PDT experiment on LNCaP cells was realized in the presence or in the absence of an excess of M6P (10 mM). M6P was added in the culture medium in order to saturate the M6PR before cells incubation (4 h) with M6C-Man-MSN and irradiation. This experiment showed that the PDT effect was totally inhibited by the presence of an excess of M6P and proved the involvement of CI-M6PR in this mechanism.

In addition, the fluorescent imaging of LNCaP cells showed the high colocalization of M6C-Man-MSN with lysosomes as expected, because CI-M6PR is involved in the routing to lysosomes (Figure 3b). Finally, a PDT experiment was realized on human fibroblasts known to express a normal level of CI-M6PR [14] in comparison to LNCaP, expressing a high level of CI-M6PR. As shown in Figure 3c, M6C-Man-MSN induced a rapid and strong PDT effect on prostate cancer cells overexpressing CI-M6PR, causing 78% cell death after 6 h of incubation. In contrast, fibroblasts presented only 30% cell death under the same conditions. After 18 h of incubation time, M6C-Man-MSN induced 52% and 85% cell death on fibroblasts and on prostate cancer cells, respectively.

## 3. Materials and Methods

### 3.1. In Vitro Photodynamic Therapy (PDT) Experimental Settings

Prostate cancer cell line (LNCaP) and healthy fibroblasts were purchased from ATCC (American Type Culture Collection, Manassas, VA, USA). LNCaP cells were routinely maintained in Roswell Park Memorial Institute Medium (RPMI) supplemented with 10% fetal bovine serum (FBS), 1% penicillin/streptomycin (P/S), 1% sodium pyruvate, 1% hepes, and 1% glucose. Fibroblasts were maintained in Dulbecco’s Modified Eagle Medium (DMEM) supplemented with 10% FBS and 1% P/S. Cells were grown in a humidified atmosphere at 37 °C under 5% CO_2_.

For in vitro phototoxicity experiments, cells were seeded into a 96-well plate, 1000 cells per well in 100 μL of culture medium, and grown for 24 h. Then, cells were incubated during different times (3, 6, 9, or 18 h) with several batches of MSN (MSN, MSN-M6C, MSN-M6C-Man) at a final concentration of 80 μg·mL^−1^. After incubation with MSN, cells were submitted (or not) to laser irradiation (650 nm, 20 min, 11.25 J·cm^−2^). Two days after irradiation, a 3-(4,5-dimethylthiazol-2-yl)-2,5-diphenyltetrazolium bromide; Promega (MTT) assay was performed for cell death quantification. Briefly, cells were incubated with 0.5 mg·mL^−1^ MTT for 4 h to determine mitochondrial enzyme activity. Then, MTT precipitates were dissolved in 150 µl ethanol/DMSO (1:1) solution and absorbance was read at 540 nm.

To demonstrate the involvement of the CI-M6PR pathway in PDT efficiency, a PDT experiment was performed in the presence or in the absence of an excess of free M6P (10 mM) added 10 min before the nanoparticles. Then, LNCaP cells were incubated with 80 µg·mL^−1^ of MSN-M6C-Man for 4 h and irradiated. Cell death was determined by an MTT assay as described above.

### 3.2. In Vitro Fluorescence Imaging Experimental Settings

The cellular uptake experiment was performed using confocal fluorescence microscopy on living cells.

For this, LNCaP cells were seeded (10^6^ cells·cm^−2^) on culture dishes with a glass bottom (Fluorodish—World Precision Instrument, Stevenage, UK). The day after seeding, cells were incubated with 80 μg·mL^−1^ of MSNs for 24 h. After this treatment, cells were loaded with Cell Mask (Invitrogen, Cergy Pontoise, France) at 5 μg·mL^−1^, for 15 min, for membrane staining. Then, cells were washed twice and visualized. Confocal fluorescence microscopy was performed on living cells at 633 nm for nanoparticles and 561 nm for cell membranes.

For the lysosomal colocalization study, two hours before the end of the incubation, cells were incubated with LysoTracker Red DND-99 (Invitrogen) for lysosomal staining at a final concentration of 50 nM. Before visualization, cells were washed gently with culture medium. Cells were then scanned with a LIVE confocal microscope, at 633 nm and 561 nm for nanoparticles and lysosomes, respectively.

All images were obtained with a high magnification (63x/1.4 OIL DIC Plan-Apo).

## 4. Conclusions

One of the main challenges in cancer treatment nowadays is the development of a therapy that will be efficient while being selective towards the tumor. In this context, targeted medicine is a fast-growing field. The targeting of biomarkers overexpressed by tumor cells, such as the CI-M6PR in prostate cancer cells, is a promising way to achieve selectivity. The development of new ligands with a better affinity for the receptor, such as M6C-Man, presents a double advantage: an increase in cell death and a reduction of the incubation time. This higher efficiency is probably due to a faster cell uptake of the MSN-M6C-Man and their routing in endolysosomal vesicles. Furthermore, the development of multifunctional nano-objects appears to be promising as it benefits from multiple targeting [25].

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
