# Peer review of "Efficient Photodynamic Therapy of Prostate Cancer Cells through an Improved Targeting of the Cation-Independent Mannose 6-Phosphate Receptor"

_ijms, 2019, doi:10.3390/ijms20112809_

Round 1
Reviewer 1 Report
This is realy an impressive MS, which provides meaninful results. Hope medical application of your methods can be achieved one day. Great work. Congratulations!
Please correct on page 5: "(Error! Reference source not found.1)".
I judged the MS as a very valuable contribution and recommended publication.
Here are the reasons, why I did so:
1. Photodynamic therapy becomes increasing interesting in control of pathogens and cancer cells
2. Prostate cancer is one of the most frequent cancer in man. Although new therapies have decreased mortality it is still a disease leading to numerous fatalities.
3. Authors have designed an interesting TEOS-nanoparticle based porphyrin-photosensitzer, with a specific bait for prostate cancer cells attached (M6C-Man).
4. PDT is not specific, because in general it does not distinguish between cancer cells and healthy cells (as almost all other cancer therapies, except modern immune-based methods, which are about to become more and more relevant)
5. In this work the authors used a previously described mannose-6-phosphate-receptor (Angew Chem Int Ed Engl 2015 May 20;54(20):5952-6. Epub 2015 Mar 20) as a target in order to achieve specific uptake of a photosensitizer by prostate cancer cells (over expressed in many prostate cancer celllines).
6. The achievement of this cooperation was the synthesis of a photoynamic molecule with mannose as a bait.
7. Authors could clearly indicate uptake of the particle inside the cell
8. Sensitivity against a prostate cell-line compared to fibroblasts was demonstrated
Detailed review about the particular sections of the MS:
Introduction: Brief but stringent description of the present state of research, accounting relevant literature as well explanation of precursor experiments, which were the baseline for the presented research.
Material and Methods + Supplement: The synthesis steps are quite complicated and need a long description (alone the di-Mannose bait MSM-M6C-Man needs 11 synthesis steps). So the authors have just provided an overview of the methods in the MS but have provided a detailed step by step procedure of the synthesis with corresponding quality check of all intermediates. This is from interest for chemist, while researchers from medical fields and molecular science are more interested in the properties, specifity and activity of the photosensitizer, which is provided in the main MS.
Result (no discussion): Synthesis of the molecule is the first part of the MS (but fits here, although it might be moved into the MM-section). The results of invitro-experiments show clearly the effect of different mannose-ligands on cell survival after photodynamic treatment (three independent experiments). Confocal images show clear uptake of particles with mannose-bait in prostate cancer cells.
The results are very promising with respect to prostate cancer research and (by changing ligand) may also be interested in other cancer types. I very much suggest publication of this MS in order to communicate this promising achievements to the scientific community.
Author Response
We sincerely thank the reviewer 1 for his/her comments extremely encouraging and motivating. The analysis reported here is detailed, precise and underlines all the crucial points of this work.
As suggested the error p 5 was corrected and the particles inside the cells in figure 2 were highlighted by white arrows. In addition, the legend was updated.
Concerning the synthesis of the molecule, if possible, we would prefer to keep it in the main manuscript because it corresponds to an important part of the thesis work of the first author. In fact, if this compound is now easy to produce and cheap, it was at the beginning a strong challenge and the development of the operating mode was tedious and time-consuming. This molecule is the heart of the prostate cancer cells targeting and the key of the PDT efficiency. It is important for us to valorize its synthesis by a chapter in the main manuscript. For this reason we hope that the referee will agree to keep this part in the manuscript.
Sincerely yours
Magali Gary-Bobo
Reviewer 2 Report
In the article entitled “Efficient photodynamic therapy of prostate cancer cells through an improved targeting of the cation independent mannose 6-phosphate receptor” by Alain Morère and co-workers, the authors developed novel compounds that allows the improvement of the targeting of the cation-independent mannose 6-phosphate receptor, overexpressed in prostate cancer. The original sugar, a dimannoside-carboxylate (M6C-Man), grafted on the surface of MSN bearing a porphyrin for PDT applications, leads to a higher endocytosis and thus increases the efficacy of MSN.
This is an interesting article, in general presented in a clear way and with very motivating achievements.
However before being accepted the authors must consider the following comments:
My first concern is related with the porphyrin used. How it was prepared? When I consult the reference Int. J. Pharm. 2012, 423, (2), 509-15 indicated, I could not find the porphyrin. So if this porphyrin is new, its characterization must be added or the reference adequately indicated.
In scheme 2 correct the formula of reagents putting the numbers as subscript.
Indicate in schemes what is TMSCl as it was done for DMP
The authors must be coherent concerning the use of bold to indicate abbreviations (e.g. M6C).
In SI the structures must be numbered (can be partially) in order to facilitate the identification of the signals.
in SI the authors must revise the NMR since some multiplets are just indicated by a number and a range of values must be given (see for example Compound 1 but there are others).
In the chemical shifts sometimes there is a space between the number and the unit. Protons coupling with each other must have the same coupling constant (see for example compound 6). Most are right but it is possible to find some inconsistency.
In page 8, lines 7 and 8 in the sentence #al concentration of 80 μg.mL-1. After incubation with MSN, cells were submitted (or not) to laser irradiation (650 nm, 20 min, 11.25 JNaN-2)# indicate the units correctly
In SI the authors refer that the Silylation procedure of the porphyrin was performed according with reference 1 and no reference was indicated in S.I
No reference to the instruments used to characterize the nanoparticles was indicated
In references the DOI of the articles were not mentioned
How the authors compare these results with the previous ones obtained by the group?
Author Response
We really thank the reviewer 2 for his/her careful reviewing of the manuscript and for his/her helpful and motivating comments.
Below are listed the answers to the points:
“My first concern is related with the porphyrin used. How it was prepared? When I consult the reference Int. J. Pharm. 2012, 423, (2), 509-15 indicated, I could not find the porphyrin. So if this porphyrin is new, its characterization must be added or the reference adequately indicated.”
This mistake was corrected. The synthesis of the porphyrin was described in Int. J. Pharm. 2010, 402, (1-2), 221-230 and this reference is now adequately indicated.
“In scheme 2 correct the formula of reagents putting the numbers as subscript.”
The numbers were put in subscript in the formula of reagents
“Indicate in schemes what is TMSCl as it was done for DMP “
The name Trimethylsilyl for TMS was written in schemes.
“The authors must be coherent concerning the use of bold to indicate abbreviations (e.g. M6C). “ All abbreviations were written in bold (i.e. M6C, M6C-Man, MSN)
“In SI the structures must be numbered (can be partially) in order to facilitate the identification of the signals. “
As required the structures were numbered in SI.
“in SI the authors must revise the NMR since some multiplets are just indicated by a number and a range of values must be given (see for example Compound 1 but there are others). “
Range of values are now given for each multiplet and it has been carefully corrected for each molecules.
“In the chemical shifts sometimes there is a space between the number and the unit. Protons coupling with each other must have the same coupling constant (see for example compound 6). Most are right but it is possible to find some inconsistency. “
The space has been removed between number and unit. The coupling constants have been carefully checked for each compound.
“In page 8, lines 7 and 8 in the sentence #al concentration of 80 μg.mL-1. After incubation with MSN, cells were submitted (or not) to laser irradiation (650 nm, 20 min, 11.25 JNaN-2)# indicate the units correctly “
This was corrected, you will find 11.25 JNaN-2
“In SI the authors refer that the Silylation procedure of the porphyrin was performed according with reference 1 and no reference was indicated in S.I “
The reference Int. J. Pharm. 2010, 402, (1-2), 221-230 is now indicated.
“No reference to the instruments used to characterize the nanoparticles was indicated”
In fact, this chapter was lacking, the paragraph below concerning the characterization of nanoparticles was added:
TEM analysis was performed on a JEOL 1200 EXII instrument. Dynamic light scattering analyses were performed using a Cordouan Technologies DL 135 Particle size analyzer instrument. N2 adsorption isotherms were measured using a TRISTAR 3000 gas adsorption analyzer instrument. The specific surface area was determined using the BET method. UV‐vis absorption spectra were recorded on a Hewlett‐Packard 8453 spectrophotometer and the correction factors used here are those supplied by the manufacturer.
In references the DOI of the articles were not mentioned
All DOI were added when available
“How the authors compare these results with the previous ones obtained by the group?”
In previous paper (Angew. Chem. Int. Ed. Engl. 2015, 54, (20), 5952-6) we demonstrated that a mannose 6-carboxylate bioistosteric of the M6P was very efficient to allowed high photodynamic therapy of prostate cancer cells that overexpressed the CI-M6P receptor. As described in this paper we compared the affinity of the disaccharide mannose 6-carboxylate with the monosaccharide mannose 6-carboxylate and we demonstrated the higher efficiency of the disaccharide.
Sincerely yours
Magali Gary-Bobo
Round 2
Reviewer 2 Report
I am satisfied with the corrections performed by the auhors.